# How Should We Evaluate LLM Reasoning Quality For Fact Verification?

## Abstract

The reasoning traces generated by Large Language Models (LLMs) are increasingly used to improve final predictions, enable reinforcement learning based on reasoning trace correctness, and justify model outputs to users. Their recognized utility spurred a line of works on evaluating LLM reasoning quality. However, such current reasoning evaluation methods are typically generic and do not shed light on the different reasoning types that may be required for various complex tasks. In this paper, we investigate reasoning quality for the prominent task of *Fact Verification*, where a model should determine whether a given claim is entailed by a reference source text, a fundamental process known as Natural Language Inference (NLI). Specifically, we propose a novel evaluation framework that considers the prominent types of inference steps involved in NLI reasoning: hypothesis *decomposition* into individual facts, followed by source *attribution* and *entailment* decision for each fact, and finally *aggregation* of fact level decisions into the final entailment classification. Our protocol introduces fine-grained metrics to assess both the existence (whether a step was performed) and the quality (how well it was performed) for each inference type. Following this framework, we first conduct a meticulous manual evaluation of six prominent LLMs, and then scale the evaluation using LLM-as-a-Judge. Our analysis reveals several insights, including: (1) a significant positive correlation exists between the quality of the reasoning trace and the correctness of the final prediction; (2) models often omit necessary reasoning steps, leading to incomplete justifications; and (3) guiding the LLM towards a systematic reasoning trace based on our framework often improves the quality of both the reasoning trace and the overall entailment classification, specifically for "non-reasoning" models. Overall, our work provides a more diagnostic and nuanced approach to understanding and evaluating LLM reasoning trace, demonstrated specifically for NLI reasoning in fact verification, proposing insights for future improvements in reasoning quality and its downstream usage.

## 1 Introduction

Explicit reasoning traces have important benefits like improving final answer prediction (Wei et al., 2022; Sprague et al., 2025); guiding search algorithms over potential reasoning paths to improve model predictions (Hao et al., 2024; Sun et al., 2024); providing granular feedback for fine-tuning models via reinforcement learning, where the reward is based on the quality of intermediate steps (Lai et al., 2024; Lu et al., 2024; Sun et al., 2024); and might serve as a verification and justification of the final predictions in human-AI collaboration scenarios (Barez et al., 2025).

The growing importance of these reasoning traces has spurred research into methods for evaluating their quality. Existing approaches typically fall into two categories. Some provide a single, holistic score for an entire reasoning trace (Saparov & He, 2023; He et al., 2024), while others adopt a more granular, step-by-step analysis, evaluating generic properties of inference steps, such as groundedness or faithfulness (Prasad et al., 2023; Li et al., 2025). However, a key limitation of these step-by-step methods is that they often apply a uniform set of criteria to every step, disregarding the distinct functional role that each type of step plays within a larger, structured reasoning process (like those elaborated below for NLI reasoning).

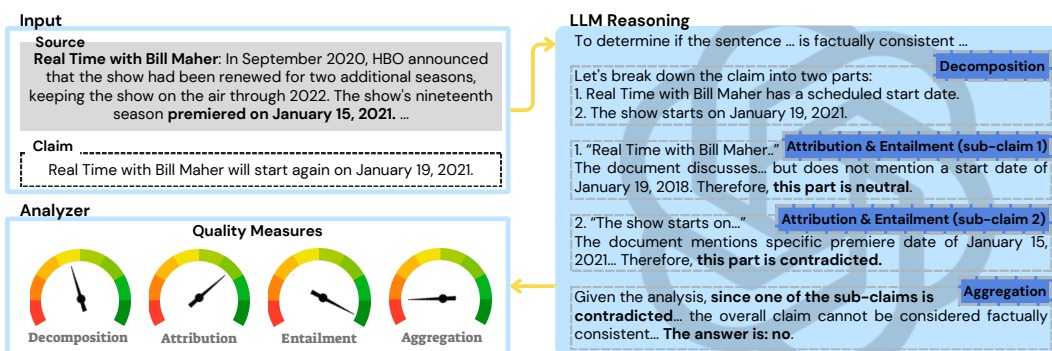

Figure 1: Overview of our evaluation framework. A model produces a reasoning trace before its final prediction on the verification of a given claim. Our framework then evaluates the quality of this trace across four distinct components: Decomposition, Attribution, Entailment, and Aggregation.

This limitation is particularly salient for complex, multi-stage tasks. In this work, we focus on entailment classification, specifically in the prominent context of fact verification, as often applied either to LLM- or human-generated claims (Tian et al., 2020; Thorat et al., 2025; Ádám Kovács & Recski, 2025; Paudel et al., 2025). The underlying reasoning required in this setting is Natural Language Inference (NLI, aka as textual entailment), where a model should determine if a reference text (*premise*) entails a given claim (*hypothesis*) (Dagan et al., 2005; Bowman et al., 2015). A robust reasoning process for NLI typically involves several distinct types of reasoning steps: decomposing the claim into sub-claims, attributing each sub-claim to supporting or refuting evidence in the source, classifying the entailment status of each sub-claim, and finally, aggregating these individual judgments into a final verdict for the entire original claim (see Fig. 1).

Building on this structured view of NLI reasoning, we introduce a novel evaluation protocol designed to assess a model's proficiency at each distinct type of reasoning steps. Our scheme evaluates each component – from decomposition to aggregation – along two axes: its *existence* (i.e., whether the model included the step in the reasoning trace) and its *quality* (e.g., the correctness of the step). This protocol is designed for both human evaluation as well as for scalable automated evaluation using an LLM-as-a-Judge.

Our experiments reveal that models often fail to produce a complete and valid reasoning trace, and often omit some necessary steps. Often, guiding the LLM to generate a complete reasoning trace, consisting of all required inference stages, successfully encourages this behavior, and in some cases improves the quality of both reasoning and final task accuracy. Furthermore, our analysis confirms a substantial correlation between the quality of individual reasoning steps and the final prediction. Specifically, the correctness of the Attribution and Entailment components for each sub-claim are strong predictors of a correct final answer, increasing confidence in the model's conclusion when these steps are performed correctly.

Overall, our contributions include: (1) a novel evaluation methodology for NLI reasoning, that decomposes the complex, multi-stage task of fact verification into distinct, functionally-motivated components; (2) a set of fine-grained metrics for evaluating both the existence and the quality of each reasoning component, enabling a more precise diagnosis of model performance; (3) an extensive empirical analysis of six prominent LLMs, with both unguided and guided prompting, providing a detailed comparison of their capabilities at each stage of the NLI reasoning process.

## 2 BACKGROUND

**Fact Verification.** Fact verification was first formally defined as a computational task by Vlachos & Riedel (2014), where the goal is to assess the faithfulness of a given claim. This process typically involves two main stages: retrieving a relevant evidence document and then verifying the claim against that evidence. The focus of our project is on the verification step, assuming an evidence document has already been retrieved. While the main body of fact verification research has focused on textual evidence (Wang, 2017; Thorne et al., 2018; Kamoi et al., 2023; Schuster et al., 2021),

a few studies have also built datasets with other evidence types, such as tables and images (Chen et al., 2020; Yao et al., 2023). Many approaches have been proposed for the verification step itself. A dominant paradigm is to frame the problem as an entailment decision (Dagan et al., 2005), where the system must determine if the evidence supports or refutes the claim. A few methods relied on fine-tuned models (Yang et al., 2021; Chen et al., 2022; Tang et al., 2024), while more recent approaches have leveraged LLMs (Zeng & Gao, 2023; Li et al., 2024; Parvez, 2025). We focus on this latter approach, specifically on cases of prompting LLMs to produce explicit reasoning tokens Lei et al. (2023); Wadhwa et al. (2024); Wan et al. (2025).

**Reasoning Steps Usefulness.** The role and usefulness of intermediate reasoning text, aka Chain-of-Thought tokens, is a subject of ongoing debate in the community. A significant body of work argues that chain-of-thought (CoT) tokens are not faithful explanations of a model's decision-making process; that is, they do not necessarily reflect the internal computations that produce the final answer (Kambhampati et al., 2025; Barez et al., 2025). These studies show that reasoning traces may be partially incorrect, omit critical information, or present a plausible but fabricated justification for an incorrect prediction (Turpin et al., 2023; Stechly et al., 2025; Bhambri et al., 2025).

At the same time, there is compelling counterevidence that higher-quality reasoning traces demon-strably improve final task performance (Liao et al., 2025; Gandhi et al., 2025). Their practical utility has been shown in several applications, such as guiding search over multiple reasoning paths or providing reward signals for fine-tuning models via reinforcement learning (Hao et al., 2024; Sun et al., 2024; Lai et al., 2024; Lu et al., 2024). This has led to the perspective that CoT should be treated as a valuable communication tool, provided its quality is systematically evaluated (Barez et al., 2025). Our work is motivated by this view: rather than demanding strict faithfulness to the model's internal process, we propose a protocol to rigorously evaluate the reasoning trace's quality for the downstream task of justifying an NLI decision.

**Reasoning Steps Evaluation.** Prior evaluations of reasoning traces proceed along three main lines. (i) *A unified score* methods score the entire reasoning process as a whole (Saparov & He, 2023; Han et al., 2024; He et al., 2024). (ii) *Reference-based* approaches compare each generated step to some gold steps (Hao et al., 2024; Li et al., 2025). (iii) *Reference-free* approaches propose generic, step-level criteria – including groundedness, semantic consistency, logical validity, fluency, minimality, and efficiency (Prasad et al., 2023; Golovneva et al., 2023; Saparov & He, 2023; Zhou et al., 2025; Qiu et al., 2025; Chen et al., 2025a; Li et al., 2025).

While flexible, the primary limitation of most of the existing reference-free metrics is their generic nature. They either treat each step in isolation or, at best, check for local consistency with the preceding steps. Crucially, they neglect the fact that steps within a complex reasoning process serve distinct functions, yet they apply a uniform set of evaluation criteria to all of them. This approach fails to provide a nuanced understanding of a model's capabilities at different stages of a task. In contrast, we argue for a type-aware evaluation. We propose to distinguish between step types and evaluate each based on its specific function. This methodology enables a more fine-grained analysis of a model's ability to execute the different kinds of reasoning steps necessary to produce a final, aggregated answer.

# 3 A PROTOTYPICAL NLI REASONING SCHEME

A prerequisite for assessing current models' entailment reasoning capabilities is understanding the expected structure of a complete reasoning process. A prominent available source for this purpose can be found in the annotation guidelines used in previous works that constructed NLI and fact verification datasets. These guidelines often instruct annotators to follow a systematic workflow before deciding on the entailment judgment of a claim. Additionally, we are inspired by established system architectures for entailment classification models, where often different components were responsible for different types of reasoning steps.

A common first step, particularly relevant in real-world scenarios where the given claims are nat-urally occurring fairly long sentences, is to *decompose* the claim into smaller sub-claims, making it easier to verify each sub-claim separately (Min et al., 2023; Kamoi et al., 2023; Mishra et al., 2024; Mitra et al., 2025). This decomposition necessitates two subsequent steps: determining the

Figure 2: Overview of a comprehensive NLI reasoning process. (i) Decomposition: the original claim is split into individual sub-claims. (ii) Attribution & Entailment: each sub-claim is checked against the source for supporting evidence, refuting evidence, or no evidence. (iii) Aggregation: if all sub-claims are supported, the claim is accepted; otherwise, it is rejected.

*entailment* status for each sub-claim (i.e., whether it is supported or not by the given source, optionally distinguishing contradiction vs neutral cases), and then an *aggregation* of these individual judgments into a final decision. The aggregation logic dictates that the entire claim is supported only if all sub-claims are entailed; otherwise, it is considered not supported. As a special case, if no decomposition occurs, the entailment decision is simply made for the claim as a whole, where no aggregation is needed.

The reasoning process must also include a meaningful step of *attribution* (i.e. evidence detection), where the model searches the source for evidence that either supports or refutes a sub-claim (Camburu et al., 2018; Niu et al., 2024; Wang & Atanasova, 2025). If such evidence is found the sub-claim entailment status is classified accordingly (entailed/supported or contradicted/refuted); otherwise the entailment status is neutral/unknown.

Finally, the correctness of a sub-claim is not always directly derivable from the evidence but may require some *inference* (Camburu et al., 2018; Bhagavatula et al., 2020; Niu et al., 2024; Havaldar et al., 2025). For example, given the sub-claim "The adult interrupted Donald Trump's speech" and the evidence "A 50-year-old man interrupted Donald Trump's speech", an inference that 'a 50-year-old man is an adult' leads to determining the entailment status. Thus, an inference step serves as a complementary bridge between attribution and the final entailment classification. The flow including the above reasoning stages is illustrated in Fig. 2.

## 4 NLI Reasoning Evaluation

The previous section outlined four key components of NLI reasoning: decomposition, attribution, entailment classification, and aggregation. Building on these, we propose several metrics that each assess a different aspect of the reasoning process. Taken together, these metrics provide a comprehensive analysis of NLI reasoning. The following metrics comprise the evaluation scheme for annotators, as detailed in Section 5.

We divide the evaluation metrics into two groups: *existence* metrics and *quality* metrics. The first group measures whether each component is present in the model's reasoning process, while the second group evaluates the quality and correctness of those components when they are executed.

### 4.1 Existence Metrics

The existence metrics are binary values that indicate, for each reasoning instance, whether a model produces each of the reasoning components: *decomposition* - whether the model decomposes the claim into smaller sub-claims; *attribution* - whether the model searches evidence for each sub-claim; *inference* - whether the model describe the inference required for an entailment classification for each sub-claim, *entailment* - whether the model determines the entailment status of each sub-claim and *aggregation* - whether the model aggregates the entailment decisions of all sub-claim (when there are a few sub-claims). The decomposition and aggregation metrics are binary values for each reasoning trace. In contrast, the other existence metrics are calculated as the proportion of sub-claims for which the component is present. Next, we describe the *quality* metrics for each component, which we only evaluate if the component exists in the reasoning trace of the respective instance.

## 4.2 DECOMPOSITION

**Granularity.**  Inspired by (Wanner et al., 2024), we define the decomposition *granularity* for a reasoning instance as the number of distinct sub-claims $\mathcal{H} = \{h_1, h_2, \ldots, h_n\}$ generated at the decomposition step. If no decomposition occurs, $\mathcal{H}$ contains a single element. The granularity score is then defined as: $G := |\mathcal{H}|$. This metric has no ground-truth value, but it can influence later steps. Low granularity leads to longer and more complex sub-claims, making attribution and entailment classification harder. High granularity increases the risk of unfaithful or incomplete decompositions. In Fig. 1 example, the granularity value is 2.

**Soundness.**  As part of the decomposition step, we assess whether the model, in its reasoning steps, generates sub-claims that are semantically entailed by the claim. The *soundness* metric measures the proportion of generated sub-claims that are consistent with the claim. The soundness score (for a reasoning instance) is defined as: $S := \frac{1}{|\mathcal{H}|} \sum_{i=1}^{|\mathcal{H}|} \mathbb{1}_{\{h_i \text{ is sound}\}}$. Intuitively, a low soundness score suggests the model introduces extraneous or fabricated sub-claims during decomposition, risking incorrect entailment judgments. In Fig. 1, both sub-claims are sound, therefore the value of this metric is 1.

**Completeness.**  For a complete view of the decomposition step, we evaluate whether the model refers all the semantic content of the original claim. The *completeness* metric checks if any part was omitted during decomposition. It is a binary value: 1 if all information is covered by the model's sub-claims, and 0 if any is missing. The completeness score is then defined as: $C := \begin{cases} 1 & \text{if } \mathcal{H} \subseteq \bigcup_i h_i \\ 0 & \text{otherwise} \end{cases}$.

Intuitively, this metric highlights cases where the model omits parts of the claim, potentially leading to incorrect predictions like falsely labeling it as *entailed*. In Fig. 1, there is no missing information, resulting in a value of 1.

## 4.3 ATTRIBUTION

**Attribution Correctness.**  The second component in a comprehensive reasoning trace begins with an attribution for each sub-claim.[1] This metric assesses whether the model correctly identifies supporting or contradicting evidence from the source, or indicates that no evidence exists, for each sub-claim. An attribution is considered correct if it justifies the sub-claim's entailment label. Formally, the metric is defined as: $A_{\text{tt}} := \frac{1}{|\mathcal{H}|} \sum_{i=1}^{|\mathcal{H}|} \mathbb{1}_{\{h_i \text{ is correctly attributed}\}}$. Intuitively, missing or incorrect attribution can cause sub-claim misclassification, leading to an incorrect overall entailment decision. For instance, in Fig. 1, the incorrect attribution for one of the two sub-claims results in a score of $1/2$.

Furthermore, we distinguish between three types of attribution: (i) **extractive** – the model copy the exact evidence span(s) from the document; (ii) **paraphrase** – the model does not extract the exact span but paraphrases the relevant content from the document; and (iii) **abstract** – the model provides a higher-level explanation of the relevant information in the document. While the specific type of attribution is not necessarily critical for the correctness of the reasoning trace, it might affect the usefulness of the reasoning trace. For example, an extractive attribution can assist a human in verifying the final answer more easily than an abstractive attribution can. Therefore, this distinction serves an analytical purpose and should not be interpreted as a measure of quality.

## 4.4 ENTAILMENT

The second phase in the *Attribution & Entailment* step is to determine the entailment status of each sub-claim. If the model has produced an attribution, it may also perform an inference step, as described in Section 3.

**Entailment Inference Correctness.**  This metric evaluates the correctness of an entailment inference, when it exists. $Infer := \frac{1}{|\mathcal{H}|} \sum_{i=1}^{|\mathcal{H}|} \mathbb{1}_{\{h_i \text{ is correctly inferred}\}}$

---

[1]If no sub-claims are generated, we treat the whole claim as a single sub-claim.

**Entailment Classification Correctness.** This metric evaluates whether the model correctly predicts the entailment label for each sub-claim by comparing the predicted label $\hat{y}_i$ with the gold label $y_i$ (provided by an oracle or human annotator).[2] $Entail := \frac{1}{|\mathcal{H}|} \sum_{i=1}^{|\mathcal{H}|} \mathbb{1}_{\{\hat{y}_i = y_i\}}$ Intuitively, misclassifying even a single sub-claim can affect the overall claim prediction, making this step crucial for reliable performance. In Fig. 1, the model makes an inference and classification decision for both sub-claims, but since the inference and the classification of the first sub-claim are incorrect, the corresponding correctness metrics are both $1/2$.

## 4.5 AGGREGATION

**Aggregation Correctness.** This metric is a binary score (0 or 1) that verifies if the model's final decision logically follows from its judgments on the sub-claims. The aggregation is considered correct if the final prediction is supported only when all sub-claims are supported, and not supported otherwise. This metric specifically captures inconsistencies between the reasoning trace and the final answer; for example, the correct aggregation in Fig. 1 earns a score of 1.

**Overall Entailment Classification.** The Overall Entailment Judgment is the final, bottom-line metric that measures the accuracy of the model's prediction against the gold-standard label, primarily used to analyze the correlation between reasoning quality and final task accuracy. A key property of our framework is its diagnostic completeness: an incorrect final prediction must originate from a flaw in at least one of the preceding reasoning components, ensuring the error is captured by our metrics. For example, the correct final answer in Fig. 1 results in a score of 1 for this metric.

## 5 EVALUATION & ANALYSIS

We begin by describing the experimental setup for our manual evaluation and then present the results of this meticulous evaluation over a representative sample, discussing key insights for both the *existence* and *quality* metrics. Following, we conduct a correlation analysis to investigate the relationships between the different reasoning components. Finally, we scale up our evaluation using a validated LLM-as-a-Judge to analyze the entire dataset, which allows us to assess a broader generalizability of our human evaluation findings.

## 5.1 EXPERIMENTAL SETUP

To analyze the reasoning traces produced by LLMs, we evaluated six prominent models: `Llama-3-1B`, `Llama-3-8B`, `Llama-3-70B`, `Gemini-2.0-Flash` `Qwen3-30B`, `Qwen3-235B`, `Grok-4-Fast`, `Gemini-2.5-Flash`, and `DeepSeek-R1-32B-Distill`. The latter two are reasoning models, which are specifically optimized for reasoning tasks[3]. For our evaluation, we randomly selected 30 samples from the recent ClearFacts dataset, which combined 14 different fact-checking benchmarks (Seo et al., 2025). We tested two prompt variants for each model: (i) an *unguided CoT prompt*, which allows the model to generate its own reasoning structure to reach a final prediction, and (ii) a *guided CoT prompt*, where we explicitly instruct the model to follow the structured three-steps reasoning process described in Section 3. The full prompts are provided in Appendix L. Decomposition examples of our system are presented in Appendix F.

The generated outputs were evaluated by three annotators all having extensive experience in similar annotation tasks. Prior to the main annotation, they completed a dedicated training session on our specific evaluation protocol. In total, our manual evaluation comprises 360 annotated reasoning instances (30 samples × 6 models × 2 prompt variants).

A detailed inter-annotator agreement analysis, described in Appendix A, shows strong results on Gwet's score (Gwet, 2008) (score range between -1 and 1): 10 of our 13 metrics achieve 'Almost Perfect' agreement, while 3 are rated as 'Substantial' – according to the common interpretation of Landis & Koch (1977). This validates the high-quality data used in our main evaluation.

---

[2] For binary classification, the *neutral* and *contradicted* classes may be merged into a single *not supported* class.

[3] For the other models, we used the non-reasoning versions.

Table 1: Manual evaluation results on the *quality metrics* across models. The results are the average scores across all the sampled instances. Results are split into (a) decomposition and attribution, and (b) entailment, aggregation, and overall correctness. Each cell shows unguided CoT (left) vs. guided CoT (right). The highest score in each column is **bold**. The two models below the horizontal line are reasoning models.

| Model | Decomposition | | | Attribution |
|---|---|---|---|---|
| | **Granularity** | **Soundness** | **Completeness** | |
| Llama-3-1B | 1.56 / 5.04 | 0.41 / 0.46 | 0.38 / 0.61 | 0.55 / 0.71 |
| Llama-3-8B | 2.32 / 2.87 | 0.97 / 0.82 | 0.93 / 0.93 | 0.97 / 0.89 |
| Llama-3-70B | 2.54 / 2.73 | 0.97 / 0.89 | 0.88 / 0.90 | 0.93 / 0.86 |
| Gemini-2.0-Flash | 2.11 / 2.33 | **1.00** / 0.96 | 0.67 / **1.00** | 0.88 / 0.90 |
| DeepSeek-R1-distill | 2.15 / 2.93 | 0.93 / 0.86 | 0.95 / 0.86 | 0.96 / 0.90 |
| Gemini-2.5-Flash | 2.20 / 3.03 | **1.00** / 0.93 | 0.88 / 0.88 | **1.00** / 0.96 |

| Model | Inference | Entailment | Aggregation | Overall |
|---|---|---|---|---|
| Llama-3-1B | 0.40 / 0.79 | 0.70 / 0.73 | **1.00** / 0.75 | 0.40 / 0.69 |
| Llama-3-8B | 0.82 / 0.81 | 0.90 / 0.83 | 0.91 / **1.00** | 0.73 / 0.75 |
| Llama-3-70B | 0.86 / 0.85 | 0.89 / 0.85 | 0.92 / **1.00** | 0.87 / **0.90** |
| Gemini-2.0-Flash | 0.80 / **0.99** | 0.75 / **0.99** | 0.89 / **1.00** | 0.77 / **0.90** |
| DeepSeek-R1-distill | 0.92 / 0.98 | 0.86 / 0.85 | 0.95 / 0.97 | 0.83 / 0.80 |
| Gemini-2.5-Flash | 0.95 / 0.96 | 0.94 / 0.96 | **1.00** / **1.00** | **0.90** / 0.87 |

## 5.2 MANUAL EVALUATION RESULTS

We begin with a manual evaluation of a representative sample of model outputs. This evaluation allows us to establish a high-quality, gold-standard dataset for our core findings.

### 5.2.1 EXISTENCE METRICS

The average results of our manual evaluation for the existence metrics, across all the models, are presented in Fig. 3. The figure details the frequency with which each reasoning component was included in the reasoning trace, comparing the performance of unguided CoT (left of each cell) and guided CoT (right of each cell) for each model. The full results are presented in Appendix B, Table 3.

We observe, from both tables, several key trends. First, models frequently omit the various types of reasoning steps when unguided. This tendency is most pronounced for the decomposition step, which models often skip entirely unless explicitly prompted. Second, model scale appears to be a factor; larger models are more likely to spontaneously perform these steps even without guidance, though this

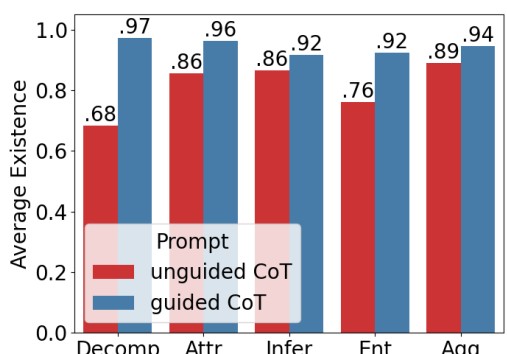

Figure 3: Manual evaluation average results on the *existence metrics* across models, grouped by unguided CoT and guided CoT.

behavior is inconsistent. Finally, the results highlight the effectiveness of guided prompting. Most models successfully adhere to the provided instructions and generate the required reasoning steps. The primary exception is the Llama-3-1B model, which struggles to consistently execute the guided steps, suggesting a potential capability threshold for following complex procedural instructions.

### 5.2.2 QUALITY METRICS

The results of our manual evaluation for the quality metrics are presented in Table 1, where the quality of a given step is evaluated for the instances where the model actually performed that step.

The manual evaluation of model performance reveals several distinct patterns. Llama-3-1B is a clear outlier, struggling to produce complete and correct reasoning traces even when guided, which correlates with its low overall prediction accuracy. While both Llama-3-8B and -70B outperform their smaller counterpart, the relationship with scale is not linear; although the 70B model achieves a 15% higher overall score, it performs worse than the 8B model on several individual reasoning components. This indicates that larger model size does not necessarily guarantee a higher-quality reasoning trace. In contrast, Gemini-2.0-Flash benefits the most from a guided reasoning prompt, showing a 13% improvement in its overall score and significant gains across most components when guided. Generally, LLMs produce high-quality reasoning traces and final classifications when guided, and reasoning models achieve strong performance even in the unguided setting.

Finally, while the specialized reasoning models generally perform well, guided prompting offers them little benefit and, in some cases, even slightly degrades their performance. Notably, when provided with guided reasoning, the performance of non-reasoning models becomes comparable to that of the reasoning models. These findings suggest that guided non-reasoning models can offer equivalent quality to reasoning models. Given that our cost analysis (Table 9) shows that non-reasoning models are cheaper and more token-efficient, we suggest that using them in a guided scenario is more efficient than using unguided reasoning models.

Additionally, four general key findings emerge from this analysis:

1. Guided prompting, while increasing the granularity of decompositions, reveals a critical trade-off between soundness and completeness. Asking for a more comprehensive breakdown improves completeness but often at the cost of soundness, as models generate subclaims not strictly entailed by the original statement. This suggests that the decomposition capabilities of current LLMs can be further improved.

2. Attribution correctness is high in the unguided reasoning setting, except for Gemini-2 and Llama-1B, though the scores are slightly lower in the guided setting. Additionally, both the inference and entailment steps achieve medium-to-high correctness (above 80% for most models) in both conditions.

3. The Aggregation step, when performed, is executed with almost-perfect accuracy across all models, meaning they almost consistently apply the correct aggregation logic.

4. With the notable exception of Llama-8B, we observe a consistent trend: for models where guided prompting enhanced the component-wise reasoning quality (e.g., Llama-1B, Gemini-2.0), there was a corresponding improvement in their overall prediction accuracy.

Further correlation analysis between generation cost, generated tokens, and other metrics, and the models' final predictions is described and illustrated in Appendix I.

### 5.3 ATTRIBUTION TYPES

A deeper analysis of the attribution types reveals a clear hierarchy in their correctness. Extractive attribution, where the model directly quotes evidence from the source, is the most reliable, with an accuracy of $93\%$. Paraphrase attribution, where the model rephrases the source text, also demonstrates high correctness at $90\%$. In stark contrast, the accuracy of abstract attribution – which requires the model to provide a high-level summary of the relevant information – drops significantly to $69\%$.

### 5.4 INTER-METRIC CORRELATIONS

To understand how the quality of one reasoning step relates to others, we computed the Spearman correlation between all quality metrics. We chose this coefficient because it is robust to outliers and non-linear relationships, making it well-suited for our medium-sized dataset. While we do not expect perfect correlations – as errors in one step can be corrected in a later one (e.g., in the Fig. 1

example, an error in one sub-claim's entailment decision does not prevent a correct final answer) – we hypothesized that a significant positive relationship exists.

The correlation matrix in Fig. 4 confirms this and reveals several key insights. The strongest correlations are between the later-stage components, with *Inference* and *Entailment* correctness being highly correlated with each other ($\rho = 0.83$) and with the *Overall* score ($\rho = 0.59$ and $\rho = 0.62$, respectively). This suggests that proficiency in these steps may be the most critical driver of a correct answer. We also observe a strong correlation between Soundness and Completeness ($\rho = 0.52$), indicating that models capable of producing sound decompositions also tend to produce complete ones. Conversely, Granularity shows a negligible correlation with all other metrics, reinforcing that simply creating more sub-claims is not an indicator of higher-quality reasoning traces. More correlation results are in Appendix C

## 5.5 LLM-AS-A-JUDGE

To scale our analysis, we employed `Grok-4-Fast` as an LLM-as-a-Judge[4], which we first evaluated for agreement against our human-annotated judgments, calculating Gwet's score between the LLM's judgments and the manual judgments (full details are in Appendix D). The LLM usually demonstrated reliable performance, achieving 'Almost Perfect' or 'Substantial' agreement on all of our 13 key metrics. We then used this judge to evaluate the reasoning traces generated for the entire ClearFacts dataset, for the same models and prompts as in the manual analysis.

The full results of this large-scale analysis are presented in Appendix D. The results for the existence metrics (see Appendix D.2, Table 6) are consistent with the trends identified in our manual analysis (Section 5.2.1), confirming that models frequently omit reasoning steps unless explicitly guided, that larger models tend to spontaneously provide the NLI reasoning components, and that guided reasoning helps the model to actually perform those steps. Additionally, some of the results for the quality metrics (from Table 1) are observed in the automatic analysis of the entire dataset (full results in Appendix D.2, Table 7), while the ranking of the models performing the reasoning trace is similar to the ranking in the manual evaluation. Additionally, although the model is worse at evaluating the inference and entailment metrics (per the lower agreement with the manual evaluation for these two metrics), it still identifies that LLMs struggle to correctly provide those components, and it identifies that the better performing reasoning components are attribution and aggregation. This suggests that improving LLMs in fact-verification may require focusing on entailment logic rather than on decomposition and attribution steps.

## 6 CONCLUSION

In this work, we proposed a novel methodology for evaluating the reasoning traces of LLMs in the context of NLI reasoning for fact verification. Our framework moves beyond generic, step-by-step metrics by decomposing the NLI reasoning process into four distinct, functionally-motivated components: decomposition, attribution, inference, and aggregation, while evaluating the existence and quality of each component separately. Our manual and automated analyses of reasoning traces shed light on specific weaknesses in current models' reasoning, such as the frequent omission of reasoning steps and the trade-offs in quality that emerge with guided prompting. This diagnostic allows for a more targeted approach to improving model reliability.

Our findings open up several avenues for future work. The high-quality reasoning traces we identified can provide the basis for extracting concise justifications that will help users verify a model's final prediction, with a guided Gemini-2.0-Flash being a particularly cost-effective choice for this task. Additionally, these traces could be used as training data to distill the reasoning abilities of large models into smaller, more efficient ones like Llama-8B or Llama-70B. Finally, our evaluation approach itself can inspire the development of similar reasoning evaluation for other prominent NLP tasks, to enable a more fine-grained analysis of their reasoning processes.

---

[4]More LLM-as-a-Judges and the agreement between them are described in detail in H

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

## A INTER-ANNOTATOR AGREEMENT (IAA)

To ensure the reliability of our annotations and establish a gold-standard evaluation set, we measured inter-annotator agreement (IAA). A subset of 30 randomly selected instances was independently annotated by all three annotators.

A key challenge of IAA measurement in this task is the frequent occurrence of highly imbalanced labels. For example, for the *Aggregation* step, nearly all instances include this component, leading to a skewed distribution. This phenomenon can cause traditional coefficients like Fleiss' Kappa to be misleadingly low despite high actual agreement (a.k.a the 'kappa paradox' (Feinstein & Cicchetti, 1990)).

To address this, we selected two complementary metrics:

1. **Gwet's AC1/AC2 Coefficient:** As our primary chance-corrected metric, we use Gwet's agreement coefficients, which are specifically designed to be robust in scenarios with skewed label distributions (Gwet, 2008; 2014). We use AC1 for binary metrics and AC2 for numerical metrics. This metric ranges from -1 to 1. We interpret the scores using the benchmarks from (Landis & Koch, 1977), where scores below 0.20 are considered 'poor', 0.21–0.40 'fair', 0.41–0.60 'moderate', 0.61–0.80 'substantial', and above 0.80 'almost perfect'.

2. **Raw Percent Agreement (RA):** For maximum transparency, and similar to

Table 2: Inter-annotator agreement results, showing Gwet's score and Raw Percent Agreement (RA) values.

| Metric | Existence | | Correctness | |
|---|---|---|---|---|
| | Gwet | RA | Gwet | RA |
| **Decomposition** | | | | |
| Decomposition | 0.66 | 0.78 | – | – |
| Granularity (F1) | – | – | 0.81 | - |
| Soundness | – | – | 0.97 | 0.98 |
| Completeness | – | – | 0.70 | 0.76 |
| **Attribution & Entailment** | | | | |
| Attribution | 0.92 | 0.96 | 0.82 | 0.91 |
| Inference | 0.96 | 0.96 | 0.87 | 0.90 |
| Entailment | 0.84 | 0.87 | 0.75 | 0.81 |
| **Aggregation** | | | | |
| Aggregation | 0.92 | 0.93 | 0.87 | 0.89 |
| Overall Decision | – | – | 0.89 | 0.91 |

Table 3: Performance of the existence metrics of NLI reasoning across models annotated by the manual analysis. Each cell shows unguided CoT (left) vs. guided CoT (right). Highest values are in bold.

| Model | Decomposition | Attribution | Inference | Entailment | Aggregation |
|---|---|---|---|---|---|
| Llama-3-1B | 0.57 / 0.97 | 0.47 / 0.80 | 0.62 / 0.65 | 0.44 / 0.67 | 0.54 / 0.77 |
| Llama-3-8B | 0.93 / **1.00** | 0.83 / 0.97 | 0.89 / 0.92 | 0.67 / 0.94 | 0.88 / 0.93 |
| Llama-3-70B | 0.80 / **1.00** | 0.98 / **1.00** | 0.90 / **1.00** | 0.87 / **1.00** | **1.00** / 1.00 |
| Gemini-2.0-Flash | 0.30 / 0.90 | 0.97 / **1.00** | 0.83 / 0.98 | 0.87 / 0.98 | **1.00** / **1.00** |
| DeepSeek-R1-distill | 0.67 / 0.97 | **1.00** / **1.00** | 0.98 / 0.95 | 0.92 / 0.98 | **1.00** / **1.00** |
| Gemini-2.5-Flash | 0.83 / **1.00** | 0.88 / **1.00** | 0.96 / **1.00** | 0.80 / 0.97 | 0.92 / 0.97 |

(Wu et al., 2025), we also report Raw Agreement. This intuitive metric measures the proportion of agreeing pairs among all possible rater pairs for a given item, averaged across all items.

Since the metrics for Soundness, Attribution, and Entailment are evaluated on a per-sub-claim basis, and annotators might identify different sets of sub-claims from the same unstructured LLM output, we first established a consistent basis for comparison. We performed a pairwise alignment of the sub-claims extracted by each annotator and calculated the *F1-Score* to measure the consistency of their decompositions. This score, which balances precision and recall between the sets of extracted sub-claims (calculated as the number of sub-claims identified by both annotators, divided by the total number of sub-claims identified by annotator A; and vice versa), is reported as Granularity (F1) in Table 2. With this alignment established, we then calculated pairwise agreement for all other metrics and averaged the results across the three pairs of annotators (A-B, B-C, A-C).

The results demonstrate a high degree of reliability in our annotation scheme. Across the 12 core quality and existence metrics, the average chance-corrected agreement (Gwet's score) is 0.85 ('Almost Perfect'), and the average Raw Agreement is 0.89. A more detailed breakdown shows that 10 metrics achieved 'Almost Perfect' agreement, with the remaining 3 rated as 'Substantial', providing strong evidence for the validity of our data. The full results are available in Table 2.

We did observe that the lowest agreement scores were for the existence of the Decomposition and its Completeness. We hypothesize that this stems from the often unstructured and ambiguous nature of the LLM's output, which can make the identification of distinct sub-claims subjective. This was particularly true in the unguided CoT setting, where varied reasoning traces corresponded to lower IAA. In contrast, the uniform structures produced by guided prompts led to markedly higher agreement, though this trend was not statistically significant given our limited sample size.

## B  MANUAL ANALYSIS RESULTS

The full setup of the manual analysis is described in Section 5. The results for each model for each prompt, for the existence metrics are presented in Table 3. The results for the quality metrics are presented in Table 1.

## C  CORRELATION WITH FINAL PREDICTION

The correlation metrics between all the quality metrics are presented in Fig. 4 and described in Section 5.4.

Table 4: Correlation of reasoning metrics with final prediction correctness. Values are point-biserial correlation coefficients ($r_{pb}$) with associated $p$-values. '(?)' indicates an existence metric, while '(#)' indicates a correctness metric. The two strongest correlates are highlighted in bold.

| Metric | $r_{pb}$ | $p$-value |
|---|---|---|
| Decomposition (?) | 0.11 | 0.045 |
| Granularity (#) | -0.06 | 0.283 |
| Soundness (#) | 0.21 | $8.0 \times 10^{-5}$ |
| Completeness (#) | 0.22 | $2.5 \times 10^{-5}$ |
| Attribution (?) | 0.18 | $4.6 \times 10^{-4}$ |
| Attribution (#) | 0.28 | $1.9 \times 10^{-7}$ |
| Inference (?) | 0.20 | $1.3 \times 10^{-4}$ |
| Entailment (?) | 0.25 | $1.3 \times 10^{-6}$ |
| **Inference (#)** | **0.44** | $2.7 \times 10^{-16}$ |

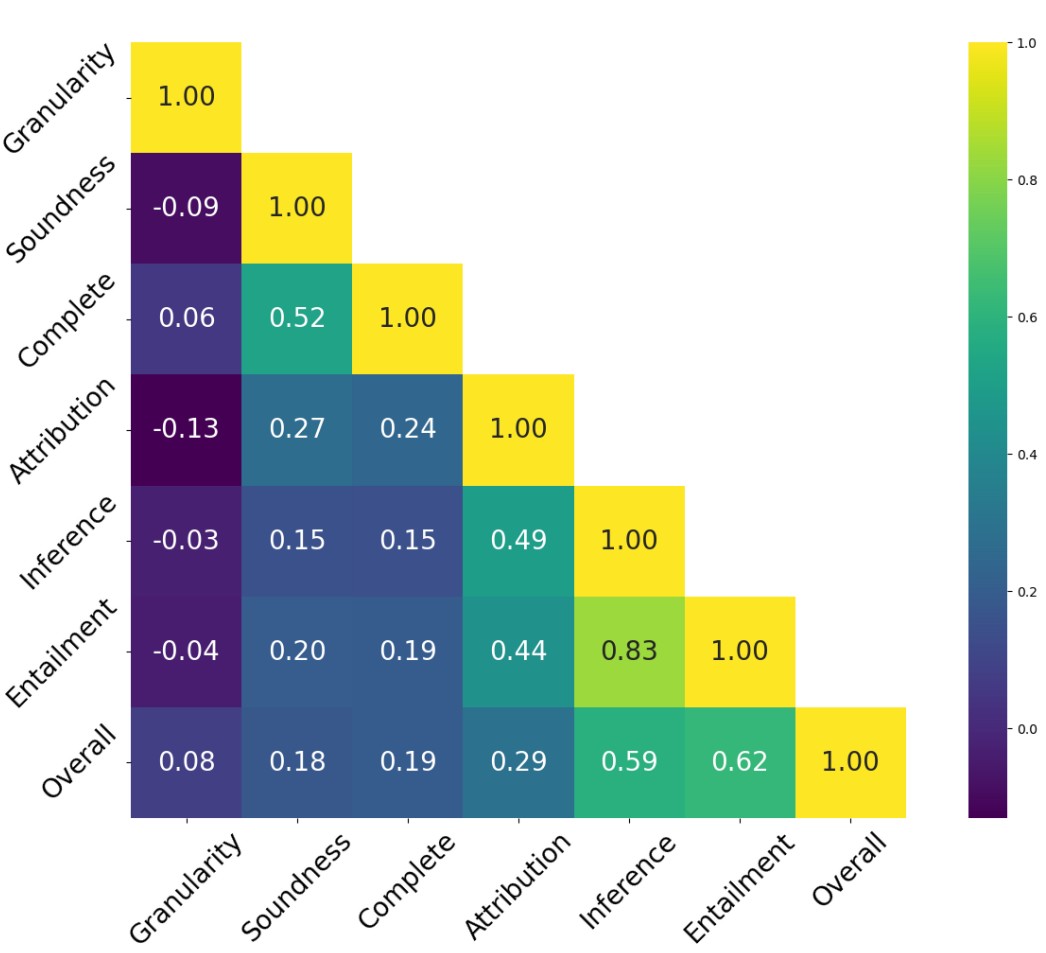

Figure 4: Spearman correlation matrix for the reasoning quality metrics. Yellow indicates a strong positive correlation, while dark purple indicates a weak or no correlation.

To understand which reasoning steps correlate with the *overall* NLI decision, we conducted a correlation analysis. We measured the relationship between each metric's value and the final binary outcome (i.e., whether the model's overall prediction was correct). Given the binary nature of the outcome variable and the continuous nature of our metric scores, we used the *Point-Biserial correlation coefficient* ($r_{pb}$; Tate (1954)). In this setup, we investigate the relationship between the performance on each reasoning component and the final answer's correctness.

The analysis, presented in Table 4, yields a crucial high-level insight: with the sole exception of *granularity*, all of our proposed reasoning metrics show a statistically significant, positive correlation with the final prediction's correctness. This demonstrates that the structured reasoning process, when performed correctly, is not merely superfluous text; it serves as a reliable witness to the validity of the final answer. The presence and quality of these intermediate steps are meaningfully linked to the model's success on the task.

Drilling deeper into these correlations, the results reveal a clear hierarchy of importance among the reasoning components. The strongest predictors of a correct final answer are, by a significant margin, the metrics evaluating the correctness of the entailment classification step. Specifically, *entailment classification correctness* shows the highest correlation ($r_{pb} = 0.55, p < 0.001$), followed by *entailment inference correctness* ($r_{pb} = 0.44, p < 0.001$). This finding underscores the intuitive conclusion that correctly solving the individual sub-problems is the most direct path to overall success on the NLI task.

Furthermore, the results highlight that the quality of a reasoning step is a more reliable indicator of success than its mere existence. For instance, *attribution correctness* ($r_{pb} = 0.28$) has a considerably stronger correlation than *attribution existence* ($r_{pb} = 0.18$). The only metric with no statistically significant correlation is *granularity* ($p = 0.28$), indicating that the number of sub-claims a model generates has no bearing on its success.

## D  LLM-AS-A-JUDGE

### D.1  LLM-AS-A-JUDGE AGREEMENT

To further ground our insights of the quality of the reasoning steps of models in entailment decision, we use LLM-as-a-Judge to evaluate the reasoning trace of the whole dataset. We instruct a model to produce all the existence and quality metrics, similar to the annotators. Out of 13 Gwet metrics, 5 metrics are 'Almost Perfect', the rest are 'Substantial'. The full results are in Table 5.

### D.2  RESULTS

The full LLM-as-a-Judge experiment setup is described in Section 5.5. The results on the *existence* metrics are in Table 6. The results on the *quality* metrics are presented in Section 5. The differences in the observations between the manual and the automated evaluations are described in Section 5.5.

Table 5: Agreement between LLM-as-a-Judge and the evaluation set. Each cell shows the Judge value.

| Metric | Existence | | Quality | |
|---|---|---|---|---|
| | Gwet | RA | Gwet | RA |
| **Decomposition** | | | | |
| Decomposition | 0.69 | 0.81 | – | – |
| Granularity | – | – | 0.87 | 0.88 |
| Soundness | – | – | 0.78 | 0.81 |
| Completeness | – | – | 0.87 | 0.89 |
| **Attribution & Entailment** | | | | |
| Attribution | 0.81 | 0.82 | 0.71 | 0.74 |
| Inference | 0.69 | 0.72 | 0.65 | 0.69 |
| Entailment | 0.73 | 0.76 | 0.64 | 0.68 |
| **Aggregation** | | | | |
| Aggregation | 0.90 | 0.91 | 0.84 | 0.87 |
| Overall Decision | – | – | 0.78 | 0.85 |

Table 6: LLM-as-a-Judge analysis results of the *existence metrics* across models. Each cell shows unguided CoT (left) vs. guided CoT (right), rounded to two decimals. Highest scores in **bold**.

| Model | Decomposition | Attribution | Inference | Entailment | Aggregation |
|---|---|---|---|---|---|
| Llama-3-1B | 0.09 / 0.67 | 0.71 / 0.74 | 0.56 / 0.51 | 0.85 / 0.83 | 0.90 / 0.81 |
| Llama-3-8B | 0.54 / 0.98 | 0.97 / 0.96 | 0.87 / 0.83 | 0.95 / 0.99 | 0.92 / 0.93 |
| Llama-3-70B | 0.60 / **0.99** | 0.99 / 0.98 | 0.93 / 0.89 | 0.99 / **1.00** | **1.00 / 1.00** |
| Gemini-2.0-Flash | 0.16 / 0.86 | **1.00** / 0.98 | 0.93 / 0.91 | 0.99 / **1.00** | **1.00 / 1.00** |
| Qwen3-30B | 0.56 / 0.91 | 0.99 / 0.99 | 0.94 / 0.94 | **1.00 / 1.00** | **1.00 / 1.00** |
| Qwen3-235B | 0.64 / 0.95 | 0.99 / 0.99 | 0.93 / 0.93 | **1.00 / 1.00** | **1.00 / 1.00** |
| Grok-4-Fast | 0.31 / 0.98 | 0.65 / 0.98 | 0.62 / 0.93 | 0.94 / **1.00** | **1.00 / 1.00** |
| DeepSeek-R1-Distill | 0.35 / 0.93 | **1.00** / 0.99 | **0.96** / 0.91 | **1.00** / 0.99 | **1.00 / 1.00** |
| Gemini-2.5-Flash | 0.45 / 0.97 | 0.96 / 0.99 | 0.92 / 0.91 | **1.00 / 1.00** | **1.00 / 1.00** |

Table 7: LLM-as-a-Judge analysis results on the *correctness metrics*. Each cell shows unguided CoT (left) vs. guided CoT (right), rounded to two decimals. Highest values in each column are in **bold**.

| Model | Granularity | Soundness | Completeness | Attribution |
|---|---|---|---|---|
| Llama-3-1B | 3.31 / 3.31 | 0.82 / 0.58 | 0.59 / 0.59 | 0.55 / 0.63 |
| Llama-3-8B | 2.67 / 2.96 | 0.98 / 0.88 | 0.91 / 0.88 | 0.89 / 0.85 |
| Llama-3-70B | 2.84 / 2.96 | 0.99 / 0.97 | 0.97 / 0.97 | 0.97 / 0.96 |
| Gemini-2.0-Flash | 2.65 / 2.58 | **1.00** / 0.97 | 0.94 / 0.98 | 0.93 / 0.95 |
| Qwen-30B | 2.77 / 2.79 | **1.00** / 0.97 | 0.99 / 0.98 | 0.97 / 0.96 |
| Qwen-235B | 2.92 / 2.98 | **1.00** / 0.97 | 0.99 / 0.99 | 0.98 / 0.98 |
| Grok-4-Fast | 2.91 / 3.08 | **1.00** / 0.97 | **1.00** / 0.99 | 0.98 / 0.98 |
| DeepSeek-R1-distill | 2.88 / 2.99 | 0.99 / 0.99 | 0.98 / 0.98 | 0.96 / 0.97 |
| Gemini-2.5-Flash | 2.84 / 3.42 | **1.00** / 0.99 | 0.97 / **1.00** | 0.98 / **0.99** |

| Model | Inference | Entailment | Aggregation | Overall |
|---|---|---|---|---|
| Llama-3-1B | 0.51 / 0.65 | 0.59 / 0.57 | 0.88 / 0.92 | 0.50 / 0.45 |
| Llama-3-8B | 0.79 / 0.78 | 0.82 / 0.77 | 0.91 / 0.91 | 0.71 / 0.70 |
| Llama-3-70B | 0.91 / 0.87 | 0.94 / 0.92 | 0.94 / 0.98 | 0.90 / 0.92 |
| Gemini-2.0-Flash | 0.89 / 0.89 | 0.87 / 0.91 | 0.98 / 0.99 | 0.86 / 0.91 |
| Qwen-30B | 0.91 / 0.89 | 0.95 / 0.93 | 0.99 / **1.00** | 0.93 / 0.93 |
| Qwen-235B | 0.93 / 0.90 | 0.96 / 0.96 | 0.99 / **1.00** | 0.95 / 0.96 |
| Grok-4-Fast | 0.91 / 0.90 | 0.92 / 0.95 | 0.99 / **1.00** | 0.90 / 0.94 |
| DeepSeek-R1-distill | 0.92 / 0.90 | 0.90 / 0.94 | 0.99 / 0.99 | 0.92 / 0.93 |
| Gemini-2.5-Flash | **0.97** / 0.90 | 0.95 / **0.98** | 0.99 / **1.00** | 0.86 / **0.98** |

When we group the agreement data by prompt type, a gap in a few metrics becomes distinguishable. The agreement on the existence of decomposition in the unguided CoT setting was low (a Gwet's score of 0.41), compared to a near-perfect score of 0.98 in the guided setting. A similar gap was observed for granularity, with a Gwet's score of 0.75 for unguided prompts versus 0.96 for guided prompts. The full results are in Table 8.

# E  NUMBER OF TOKENS AND COST

An analysis of the computational cost, presented in Table 9, reveals important efficiency trade-offs. Excluding Llama-1B, we find that non-reasoning models generally produce fewer output tokens than their reasoning-focused counterparts, even in the guided scenario. However, the average cost for a guided non-reasoning model is substantially lower than for an unguided reasoning model. Given that the quality of the reasoning and the overall performance are equivalent between these two configurations (as established in Section 5.2), our findings suggest that using guided prompting with non-reasoning models is a more efficient and cost-effective strategy.

Table 8: Comparison between unguided and guided CoT prompts for the agreement between the LLM-as-a-Judge and the human annotations. AC = Gwet's AC1, RA = Raw Agreement.

| Metric | Existence | | | | Correctness | | | |
|---|---|---|---|---|---|---|---|---|
| | Unguided | | Guided | | Unguided | | Guided | |
| | AC | RA | AC | RA | AC | RA | AC | RA |
| **Decomposition** | | | | | | | | |
| Decomposition | 0.41 | 0.70 | 0.98 | 0.98 | – | – | – | – |
| Atomicity | – | – | – | – | 0.75 | 0.78 | 0.96 | 0.97 |
| Soundness | – | – | – | – | 0.62 | 0.67 | 0.71 | 0.74 |
| Completeness | – | – | – | – | 0.84 | 0.87 | 0.78 | 0.83 |
| **Attribution & Entailment** | | | | | | | | |
| Attribution | 0.67 | 0.71 | 0.87 | 0.88 | 0.60 | 0.65 | 0.65 | 0.69 |
| Inference | 0.60 | 0.65 | 0.83 | 0.84 | 0.54 | 0.60 | 0.57 | 0.62 |
| Entailment | 0.61 | 0.66 | 0.87 | 0.88 | 0.55 | 0.60 | 0.61 | 0.65 |
| **Aggregation** | | | | | | | | |
| Aggregation | 0.83 | 0.86 | 0.94 | 0.95 | 0.76 | 0.82 | 0.78 | 0.84 |
| Overall Decision | – | – | – | – | 0.65 | 0.79 | 0.55 | 0.72 |

Table 9: Average output tokens and cost across models. Results are reported for unguided CoT and guided CoT. Costs are based on the current price of the models in the OpenRouter supplier, in US dollars.

| Model | Unguided CoT | | Guided CoT | |
|---|---|---|---|---|
| | **Output Tokens** | **Cost ($\times 10^{-3}$\$)** | **Output Tokens** | **Cost ($\times 10^{-3}$\$)** |
| Llama-3-1B | 312.7 | 0.003 | 844.5 | 0.008 |
| Llama-3-8B | 243.9 | 0.007 | 346.9 | 0.010 |
| Llamam-3-70B | 285.5 | 0.010 | 329.3 | 0.012 |
| Gemini-2.0-Flash | 121.6 | 0.049 | 289.0 | 0.116 |
| Qwen3-30B | 407.9 | 0.135 | 516.4 | 0.170 |
| Qwen3-235B | 438.2 | 0.241 | 544.7 | 0.299 |
| Grok-4-Fast | 177.5 | 0.089 | 348.5 | 0.174 |
| DeepSeek-R1-distill | 443.9 | 0.120 | 548.8 | 0.148 |
| Gemini-2.5-Flash | 247.4 | 0.619 | 1398.1 | 3.495 |

# F  DECOMPOSITION EXAMPLES

Examples of the statement decompositions produced by `Gemini-2.5-Flash` are presented in Table 10. We observe that there is no single consistent structure for the decomposition, and many of the resulting sub-claims are not atomic. This suggests that the decomposition step enables further investigation.

Table 10: Decomposed Claims and Sub-claims Used in Evaluation

| Statement | Sub-claims |
|---|---|
| James Davis, the candidate of the Green Party, received the lowest number of votes in the 2016 Brisbane City Council election for Northgate Ward (a Brisbane City Council ward covering Northgate, Banyo, Nudgee, Nudgee Beach, Nundah, Virginia, Wavell Heights, and parts of Chermside and Kedron). | • James Davis was the candidate of the Green Party in the 2016 election.
• James Davis received the lowest number of votes in the 2016 Brisbane City Council election.
• The election was the 2016 Brisbane City Council election for Northgate Ward.
• Northgate Ward is a Brisbane City Council ward.
• Northgate Ward covers Northgate, Banyo, Nudgee, Nudgee Beach, Nundah, Virginia, Wavell Heights, and parts of Chermside and Kedron. |
| The United States Army National Guard 38th Sustainment Brigade has been active since 2007 and is headquartered in Kokomo, Indiana. | • The unit is the "United States Army National Guard 38th Sustainment Brigade".
• The unit has been active since 2007.
• The unit is headquartered in Kokomo, Indiana. |
| The Solheim Cup was hosted in the United States seven times between 1990 and 2013. | • The Solheim Cup was hosted in the United States seven times between 1990 and 2013.
• The hosting locations were in the "United States".
• The number of times hosted in the USA was "seven".
• The time frame for these events is "between 1990 and 2013". |
| If the thorn is deeply embedded, or if the animal is a nervous one which cannot tolerate being handled too much, then a veterinarian may be needed to remove it. | • If the thorn is deeply embedded, then a veterinarian may be needed to remove it.
• If the animal is a nervous one which cannot tolerate being handled too much, then a veterinarian may be needed to remove it. |
| Birth-weight is positively associated with breast cancer. | • Birth-weight is associated with breast cancer.
• positively associated |
| In 2013, Gaetz announced that in 2016 he would run for the 1st District state senate seat then held by his father, state senator Don Gaetz, who was due to be term-limited out of the Senate in 2016. | • In 2013, Gaetz announced.
• ...that in 2016 he would run.
• ...for a state senate seat.
• ...then held by his father, state senator Don Gaetz.
• ...who was due to be term-limited out of the Senate in 2016.
• The seat was the "1st District state senate seat". |

## G  AGGREGATION PROMPT VARIANTS

In this section, we evaluate four prompting strategies for the *aggregation* step, using four strong models. For each model and prompt variant, we report both the **aggregation score** and the **overall score**, as evaluated by the `Grok-4-Fast` LLM-as-a-Judge.

We compare the following four formulations:

Table 11: Comparison of aggregation prompt variants across four models. Scores are evaluated using `Grok-4-Fast` as an LLM-as-a-Judge.

| Model | Prompt Variant | Aggregation | Overall |
|---|---|---|---|
| Llama-3-70B | (a) Unguided CoT | 0.92 | 0.87 |
| | (b) Entailment-based | 0.79 | 0.79 |
| | (c) Contradiction-based | 0.64 | 0.67 |
| | (d) Guided CoT (Ours) | 1.00 | 0.90 |
| Gemini-2.0-Flash | (a) Unguided CoT | 0.89 | 0.77 |
| | (b) Entailment-based | 0.82 | 0.78 |
| | (c) Contradiction-based | 0.75 | 0.71 |
| | (d) Guided CoT (Ours) | 1.00 | 0.90 |
| DeepSeek-R1-Distill | (a) Unguided CoT | 0.95 | 0.83 |
| | (b) Entailment-based | 0.83 | 0.80 |
| | (c) Contradiction-based | 0.75 | 0.75 |
| | (d) Guided CoT (Ours) | 0.97 | 0.80 |
| Gemini-2.5-Flash | (a) Unguided CoT | 1.00 | 0.90 |
| | (b) Entailment-based | 0.86 | 0.83 |
| | (c) Contradiction-based | 0.80 | 0.77 |
| | (d) Guided CoT (Ours) | 1.00 | 0.87 |

(a) **Unguided CoT**: Chain-of-Though without additional guidance.

(b) **Entailment-based Aggregation** (inspired by Laban et al. (2022)): Chain-of-Though with a guidance to compute entailment scores for each sub-claim, take their mean, and classify the statement as *consistent* if the mean score exceeds $0.5$.

(c) **Contradiction-based Aggregation** (inspired by Chen et al. (2025b)): Chain-of-Though with a guidance to compute contradiction scores for each sub-claim, take their maximum, and classify the statement as *inconsistent* if the maximum score exceeds $0.5$.

(d) **Guided CoT (Ours)**: Chain-of-Thought with explicit guidance, as explained in Section 3.

**Results.** Table 11 reports the aggregation and overall scores for each model–prompt combination. Across nearly all settings, variant (d) – our guided CoT prompt – achieves the strongest aggregation performance, often with substantial improvements over alternatives.

## H  CAN AN LLM JUDGE ITSELF?

To evaluate the robustness and potential biases of LLM-as-a-Judge setups, we compute inter-annotator agreement (IAA) for three independent judges: `Gemini-2.5-Flash`, `Qwen3-235B`, and `Grok-4-Fast`. We first measure the agreement of each model with the manual annotations (see Tables 5, 12 and 13). We then examine the agreement *between* the three judges over the entire dataset, while excluding the outputs produced by those three models, in order to avoid biases (Table 14).

Next, to analyze the potential bias effects in more detail, we compute pairwise IAA scores restricted to the outputs generated by `Gemini-2.5-Flash`, serving as a representative model. These results are presented in Tables 15 to 17. Finally, we report, for each metric, the model pair achieving the highest cross-judge agreement, as summarized in Table 18. If Gemini were biased toward its own outputs, we would expect lower cross-judge agreement. Instead, Gemini–Qwen exhibits the highest agreement, suggesting LLMs might be objective judges at this setup.

## I  IMPACT OF TOKEN BUDGET, COST, AND DECOMPOSITION SIZE ON REASONING QUALITY

In this section, we examine whether several factors related to the generation process influence the quality of the NLI reasoning process. Specifically, we analyze three parameters: *(i)* the number

Table 12: IAA of Qwen3-235B-A22B-as-a-Judge with the manual annotations.

| | | Existence | | Correctness | |
| | | Gwet | RA | Gwet | RA |
| | Metric | | | | |
|---|---|---|---|---|---|
| Decomposition | Decomposition | 0.826 | 0.881 | – | – |
| | Granularity | – | – | 0.848 | 0.858 |
| | Soundness | – | – | 0.696 | 0.725 |
| | Completeness | – | – | 0.812 | 0.847 |
| Attribution & Entailment | Attribution | 0.751 | 0.767 | 0.609 | 0.647 |
| | Inference | 0.747 | 0.767 | 0.627 | 0.664 |
| | Entailment | 0.723 | 0.744 | 0.535 | 0.581 |
| Aggregation | Aggregation | 0.889 | 0.903 | 0.715 | 0.794 |
| | Overall Decision | – | – | 0.657 | 0.778 |

Table 13: IAA of Grok-4-Fast-as-a-Judge with the manual annotations.

| | | Existence | | Correctness | |
| | | Gwet | RA | Gwet | RA |
| | Metric | | | | |
|---|---|---|---|---|---|
| Decomposition | Decomposition | 0.690 | 0.811 | – | – |
| | Granularity | – | – | 0.882 | 0.891 |
| | Soundness | – | – | 0.781 | 0.805 |
| | Completeness | – | – | 0.869 | 0.891 |
| Attribution & Entailment | Attribution | 0.813 | 0.827 | 0.712 | 0.744 |
| | Inference | 0.695 | 0.724 | 0.646 | 0.685 |
| | Entailment | 0.744 | 0.769 | 0.642 | 0.682 |
| Aggregation | Aggregation | 0.899 | 0.911 | 0.842 | 0.872 |
| | Overall Decision | – | – | 0.789 | 0.858 |

Table 14: IAA between Gemini, Qwen, and Grok on all the instances, excluding the instances produced by those three models.

| | | Existence | | Correctness | |
| | | Gwet | RA | Gwet | RA |
| | Metric | | | | |
|---|---|---|---|---|---|
| Decomposition | Decomposition | 0.885 | 0.930 | – | – |
| | Granularity (F1) | – | – | 0.941 | 0.945 |
| | Soundness | – | – | 0.888 | 0.895 |
| | Completeness | – | – | 0.983 | 0.983 |
| Attribution & Entailment | Attribution | 0.852 | 0.861 | 0.792 | 0.805 |
| | Inference | 0.809 | 0.820 | 0.802 | 0.814 |
| | Entailment | 0.934 | 0.938 | 0.676 | 0.695 |
| Aggregation | Aggregation | 0.999 | 0.999 | 0.944 | 0.947 |
| | Overall Decision | – | – | 0.876 | 0.890 |

Table 15: IAA between Qwen and Grok on Gemini outputs.

|  |  | Existence | | Correctness | |
| --- | --- | --- | --- | --- | --- |
|  | Metric | Gwet | RA | Gwet | RA |
| Decomposition | Decomposition | 0.885 | 0.930 | – | – |
|  | Granularity (F1) | – | – | 0.941 | 0.945 |
|  | Soundness | – | – | 0.888 | 0.895 |
|  | Completeness | – | – | 0.983 | 0.983 |
| Attribution & Entailment | Attribution | 0.852 | 0.861 | 0.792 | 0.805 |
|  | Inference | 0.809 | 0.820 | 0.802 | 0.814 |
|  | Entailment | 0.934 | 0.938 | 0.676 | 0.695 |
| Aggregation | Aggregation | 0.999 | 0.999 | 0.944 | 0.947 |
|  | Overall Decision | – | – | 0.876 | 0.890 |

Table 16: IAA between Gemini and Qwen on Gemini outputs.

|  |  | Existence | | Correctness | |
| --- | --- | --- | --- | --- | --- |
|  | Metric | Gwet | RA | Gwet | RA |
| Decomposition | Decomposition | 0.910 | 0.945 | – | – |
|  | Granularity (F1) | – | – | 0.945 | 0.948 |
|  | Soundness | – | – | 0.872 | 0.880 |
|  | Completeness | – | – | 0.985 | 0.985 |
| Attribution & Entailment | Attribution | 0.850 | 0.859 | 0.773 | 0.787 |
|  | Inference | 0.823 | 0.834 | 0.739 | 0.755 |
|  | Entailment | 0.917 | 0.922 | 0.729 | 0.745 |
| Aggregation | Aggregation | 0.992 | 0.992 | 0.932 | 0.937 |
|  | Overall Decision | – | – | 0.831 | 0.865 |

Table 17: IAA between Gemini and Grok on Gemini outputs.

|  |  | Existence | | Correctness | |
| --- | --- | --- | --- | --- | --- |
|  | Metric | Gwet | RA | Gwet | RA |
| Decomposition | Decomposition | 0.909 | 0.946 | – | – |
|  | Granularity (F1) | – | – | 0.950 | 0.954 |
|  | Soundness | – | – | 0.882 | 0.890 |
|  | Completeness | – | – | 0.984 | 0.985 |
| Attribution & Entailment | Attribution | 0.905 | 0.911 | 0.854 | 0.863 |
|  | Inference | 0.736 | 0.753 | 0.700 | 0.719 |
|  | Entailment | 0.917 | 0.922 | 0.755 | 0.771 |
| Aggregation | Aggregation | 0.992 | 0.992 | 0.963 | 0.965 |
|  | Overall Decision | – | – | 0.812 | 0.841 |

Table 18: This table presents for each metric the pair of models with the higher LLM-as-a-Judge predcitions agreement on Gemini outputs. (a) indicating Qwen and Grok, (b) indicating Gemini and Qwen, and (c) indicating Gemini and Grok.

|  | | Existence | | Correctness | |
|  | | Gwet | RA | Gwet | RA |
|  | Metric | | | | |
| --- | --- | --- | --- | --- | --- |
| Decomposition | Decomposition | (b) | (c) | – | – |
|  | Granularity (F1) | – | – | (c) | (c) |
|  | Soundness | – | – | (a) | (a) |
|  | Completeness | – | – | (b) | (b+c) |
| Attribution & Entailment | Attribution | (c) | (c) | (c) | (c) |
|  | Inference | (b) | (b) | (a) | (a) |
|  | Entailment | (a) | (a) | (c) | (c) |
| Aggregation | Aggregation | (a) | (a) | (c) | (c) |
|  | Overall Decision | – | – | (a) | (a) |

Table 19: Regression coefficients and $p$-values for the relationship between each quality metric and the number of generated tokens, generation cost, and decomposition granularity.

| Metric | # Tokens (coeff) | # Tokens ($p$) | Cost (coeff) | Cost ($p$) | Granularity (coeff) | Granularity ($p$) |
| --- | --- | --- | --- | --- | --- | --- |
| Soundness | -9.56e-5 | 0.27 | 0.02 | 0.53 | -0.13 | 0.24 |
| Completeness | -4.35e-5 | 0.69 | 0.03 | 0.42 | 0.01 | 0.98 |
| Attribution | -1.66e-5 | 0.88 | 0.04 | 0.35 | 0.05 | 0.71 |
| Inference | 9.52e-5 | 0.42 | 0.06 | 0.13 | 0.20 | 0.18 |
| Entailment | -4.24e-6 | 0.97 | 0.04 | 0.25 | 0.03 | 0.81 |
| Aggregation | -3.21e-5 | 0.57 | 0.02 | 0.34 | -0.04 | 0.55 |
| Overall | -1.45e-7 | 0.99 | 0.06 | 0.24 | 0.04 | 0.84 |

of generated tokens, *(ii)* the monetary cost of generation, and *(iii)* the number of produced sub-claims (granularity). For each quality metric, we fit three linear regression equations, each testing the relationship between one of the generation parameters and each individual quality metric.

Table 19 summarizes the regression coefficients and corresponding $p$-values. Across all metrics, the estimated effects are small in magnitude, and the $p$-values are consistently high ($p > 0.1$), indicating no statistically significant relationship between quality and any of the examined parameters. This suggests that variations in token count, generation cost, or decomposition size do not materially affect the quality of the reasoning as measured by our evaluation framework.

# J  CLEARFACTS DATASET

We are provided the dataset composing the ClearFacts benchmark Seo et al. (2025) in Table 20.

# K  EXISTING WORK COMPARISON

Following standard evaluation protocols Golovneva et al. (2023); He et al. (2024), we calculated the correlation (Somers' D Somers (1962)) between the scores derived from our approach versus human quality judgments (1–5 Likert scale) on the e-SNLI dataset Bowman et al. (2015). The results are: ROSCOE : 0.30, SocREval (Base): 0.30, SocREval (Prompt Engineered): 0.58, Our Approach (Average of metrics): 0.45. While our method focuses on granular diagnosis rather than a single holistic quality score, it demonstrates a clear improvement over the base implementations of ROSCOE and SocREval. While heavily prompt-engineered SocREval achieves higher correlation, our method provides granular diagnostic insights that holistic scores cannot capture.

Table 20: Datasets composing the ClearFacts benchmark, along with the number of instances contributed by each dataset.

| Dataset | # Instances |
|---|---|
| AggreFact-CNN | 83 |
| AggreFact-XSum | 91 |
| ClaimVerify | 92 |
| ExpertQA | 86 |
| FactCheck-GPT | 92 |
| LFQA | 90 |
| RAGTruth | 93 |
| Reveal | 90 |
| TofuEval-MediaS | 87 |
| TofuEval-MeetB | 89 |
| Wice | 90 |
| CoverBench | 238 |
| Hover | 281 |
| SciFact | 89 |
| **Total** | **1590** |

## L  PROMPTS

We evaluated all models using two distinct prompt conditions. The *unguided CoT* condition utilized a standard chain-of-thought prompt, taken from (Wan et al., 2025), which encourages the model to produce a free-form reasoning trace. The *guided CoT* condition augmented this prompt with explicit instructions, directing the model to adhere to the specific, systematic reasoning structure defined in Section 3. The unguided CoT prompt is Prompt no. 1, and the guided CoT prompt is Prompt no. 2.

---

**Prompt 1:  NLI Unguided CoT**

Document: {document}

Sentence: {claim}

Determine if the sentence is factually consistent with the document provided above.  A sentence is factually consistent if it can be entailed (either stated or implied) by the document.
If any part of the claim is not substantiated, it should be considered inconsistent.

Let's think step by step.

Conclude your response with either "yes" (the claim is consistent) or "no" (the claim is inconsistent).

---

```
Prompt 2:   NLI Guided CoT
```

Document: {document}

Sentence: {claim}

Determine if the sentence is factually consistent with the document provided above.   A sentence is factually consistent if it can be entailed (either stated or implied) by the document.
If any part of the claim is not substantiated, it should be considered inconsistent.

Steps:
- Decompose the claim into distinct sub-claims.
- For each sub-claim:
   1.  Identify the exact text in the document that supports or contradicts it, or note that there is no relevant information.
   2. Classify the sub-claim as:
      • entailed – fully supported or implied by the document
      • contradicted – directly refuted by the document
      • neutral – neither supported nor contradicted (no evidence)
- Provide your reasoning by listing each sub-claim with its classification and evidence.
- Finally, decide:
   • "yes" if all sub-claims are entailed
   • "no" if any sub-claim is contradicted or neutral

Conclude your response with either "yes" (the claim is consistent) or "no" (the claim is inconsistent).

## M    LLM USAGE

Throughout the writing process of this paper, we utilized LLMs to assist with polishing the text, including correcting grammatical errors and improving clarity through paraphrasing.

