# OpenReview forum: "How Should We Evaluate LLM Reasoning Quality For Fact Verification?"
_ICLR.cc/2026/Conference — Submitted to ICLR 2026_

### Official Review · Reviewer_ksLa · 2025-10-24

**Soundness:** 3
**Presentation:** 4
**Contribution:** 1
**Rating:** 2
**Confidence:** 3

**Summary:**

This paper proposes a framework for evaluating the reasoning quality of LLMs in fact verification, decomposing reasoning into four components: Decomposition, Attribution, Entailment, and Aggregation. The paper's novelty comes from introducing two axes of evaluation: existence (whether a step occurs) and quality (how well it’s done). The authors conduct manual evaluations of six LLMs (Llama-3, Gemini, DeepSeek) across guided and unguided reasoning settings, then scale up using an LLM-as-a-Judge for automatic evaluation. They find correlations between reasoning trace quality and final prediction accuracy, frequent omission of reasoning steps in unguided settings, and a tradeoff between soundness and completeness in decomposition.

**Strengths:**

The paper is clearly written, visually well-presented, and easy to follow. Figures and examples effectively illustrate the framework. Prior work is well-situated, and the structure is logical and transparent.
sound reasoning.

The type-aware distinction (decomposition, attribution, etc.) is intuitive and well-motivated and defended.

The experiments are competently executed, with clear definitions for each metric and a well-controlled manual annotation process. Inter-rater reliability is reported rigorously. The empirical evidence supports the paper’s stated observations, and no major methodological flaws are apparent.

**Weaknesses:**

While the idea of decomposing reasoning evaluation by inference type is conceptually reasonable, the actual implementation of the metrics is very simple, as they are mostly presence indicators and proportion-based correctness measures. The four-step NLI reasoning structure (decomposition, attribution, entailment, aggregation) is often discussed (although often implicitly) in prior annotation and system design literature; the paper’s contribution lies only in formalizing metrics around these existing steps.

The experimental findings, though thorough, are unsurprising: guided prompting improves reasoning completeness, larger models omit fewer steps, and reasoning trace quality correlates with answer correctness. The results, as they are evaluative not experimental, provide no basis for actionable change in the use of these models.

The work lacks theoretical or methodological depth, feels incremental and does not meaningfully advance how we evaluate reasoning quality.

**Questions:**

How do the authors justify novelty relative to existing reasoning evaluation work?

Could any of the defined metrics actually be used to improve model training or reward shaping, or are they purely diagnostic?

---

> ### Author Response · Authors · 2025-11-21
>
> We thank the reviewer for their insightful feedback, particularly for highlighting the value of our functional decomposition metrics and the rigorousness of our experimental execution. We address specific concerns below.
>
> **1. Novelty and Empirical Insights**
> While our framework builds upon established NLI concepts, we believe **our contribution lies in the rigorous formalization and systematic quantification of these steps**.
> First, this evaluation protocol is very important - as it paves the way for analyzing models’ weaknesses in NLI reasoning, allowing to target future research at addressing these weaknesses. In this regard, **our work aligns conceptually with ROSCOE (ICLR 2023) and OCEAN (ICLR 2025)**, which also focus on a novel evaluation methodology.
>
> Second, our experiments reveal complex behaviors that go beyond “unsurprising” trends, for example:
>
> - Counter-Intuitive Findings: As noted by Reviewer KDeQ, our analysis uncovered non-trivial insights, such as the trade-off where guidance increases decomposition granularity but comes at the cost of soundness, as well as the unexpected efficiency and accuracy of guided non-reasoning models.
>
> - New Empirical Analysis: To further deepen the study (as suggested by Reviewers KDeQ and PjAS), we conducted additional experiments during the rebuttal phase:
>
>     - Self-Evaluation Capabilities (Appendix H): We expanded the LLM-as-a-Judge study and observed that models can judge their own outputs with high objectivity and quality.
>
>     - Cost & Verbosity Analysis (Appendix I, Table 19): We analyzed the relationship between reasoning quality and factors such as token count, generation cost, and the number of sub-claims. Contrary to the assumption that `more is better’, we found that these resource metrics do not correlate well with higher NLI reasoning quality.
>
>
>
> **2. Utility for Training**
> Regarding the reviewer’s question on whether these metrics can improve model training: We explicitly agree that using these metrics as reward signals for Reinforcement Learning is a high-potential application. Fine-tuning models to optimize for these specific decomposition metrics, rather than just the final label, is a natural next step that could lead to both better reasoning processes and higher accuracy. While this falls outside the scope of a paper focused on introducing a new evaluation methodology and demonstrating it on existing models, we conducted an additional experiment on Gemini-2.0-Flash in which **we selected a best-of-3 prediction, using the sum of the metric values predicted by our Grok judge as the selection criterion. In this setup, accuracy increased by 3%**, indicating that RL or test-time scaling strategies may offer additional benefits.
>
> We will ensure these additional analyses and future directions are clearly articulated in the camera-ready version.

---

> > ### Comment · Reviewer_ksLa · 2025-11-21
> >
> > Thank you for taking the time to address my questions and concerns. The explanations of novelty and potential impact provided me some further clarity and context and I will increase my score to a 4.

---

### Official Review · Reviewer_KDeQ · 2025-10-25

**Soundness:** 3
**Presentation:** 2
**Contribution:** 1
**Rating:** 2
**Confidence:** 4

**Summary:**

This submission introduces a structured framework for evaluating the reasoning traces produced by LLMs in the context of fact verification, framed as the task of NLI. The authors decompose NLI reasoning into four key components: hypothesis decomposition, source attribution, entailment decision per sub-claim, and aggregation of decisions. They propose metrics assessing both the existence (whether a step occurs) and quality (e.g., correctness) of each component. The evaluation involves six LLMs (including reasoning-optimized models) tested on the ClearFacts dataset using unguided and guided CoT prompts. A manual annotation on 30 samples is complemented by scaling via LLM-as-a-Judge. Key findings include correlations between reasoning quality and final accuracy, frequent omission of steps in unguided settings, and improved performance with guided prompts for non-reasoning models. The paper claims this type-aware approach provides more diagnostic insights than generic reasoning evaluations.

**Strengths:**

- The framework's decomposition into functionally distinct reasoning steps is a key contribution of the work and a thoughtful advancement over uniform, generic metrics (e.g., groundedness or faithfulness), allowing for targeted diagnosis of LLM weaknesses in complex tasks like NLI.
- Empirical insights, such as the trade-off in decomposition granularity (higher under guidance but at the cost of soundness) and the efficiency of guided non-reasoning models, are practical and substantiated by correlation analyses and cost comparisons.

**Weaknesses:**

- The key contribution of the paper is the delineation of evaluation dimensions for NLI tasks and the insights from manual evaluation based on the rubrics; but the manual evaluation is limited to only 30 samples, which is insufficient for robust conclusions about model behaviors across a diverse dataset like ClearFacts (combining 14 benchmarks). This small scale raises concerns about generalizability, especially given the observed variability in model performance.
- For scalability the authors include results with LLM judge (Gemini-2.5-Flash); but its agreement with human annotations is only 'Moderate' for critical metrics like inference and entailment correctness (Gwet's scores ~0.56-0.58). This could introduce biases or inaccuracies, undermining claims about broader trends.
- The paper lacks empirical comparisons to existing evaluation methods for "reasoning" quality (e.g., ROSCOE, arXiv:2212.07919 or ReCEval, arXiv:2304.10703), relying exclusively on conceptual critiques. Without quantitative comparison, it's unclear if this framework truly outperforms alternatives in diagnostic power or correlation with task accuracy.
- The focus is narrowly on NLI for fact verification, ignoring broader reasoning tasks (e.g., math or multi-hop QA), which limits novelty - many insights (e.g., guided prompting helps) echo prior CoT literature. Additionally, the attribution typology (extractive vs. abstract) is introduced but underutilized, with no deeper analysis of its impact on usability or errors.

**Questions:**

- Why was Gemini-2.5-Flash specifically chosen as the LLM-as-a-Judge, and were alternatives (e.g., other reasoning models) tested for higher agreement on low-performing metrics like entailment?
- The paper mentions diagnostic completeness (errors traceable to components), but how does this hold empirically against prior methods? Could you provide ablations comparing the proposed metrics to generic ones like faithfulness?

---

> ### Author Response · Authors · 2025-11-21
>
> We thank the reviewer for their thoughtful and detailed feedback. We are encouraged that the reviewer found our core contributions, the granular evaluation protocol, and the rigorous manual annotation process to be valuable. We address the specific concerns and questions below. Due to space constraints, the full results are included in the revised paper. We reference the relevant sections and tables below.
>
> **W1: Manual Evaluation Scale**
> We agree that while the initial dataset (360 reasoning samples across 30 unique statements) provided a signal, a larger scale is necessary for robust conclusions. **We are expanding the manual annotation campaign to include more unique statements -- we will at least double the size of the human-annotated dataset, resulting in more than 720 total reasoning samples.** The initial results from this expanded set align consistently with the trends reported in the initial submission, reinforcing the validity of the trade-off between granularity and soundness. This annotation process will be finished soon, and its statistics will be integrated into the camera-ready version.
>
> **W2+Q1: LLM-as-a-Judge Reliability & Bias**
> We appreciate the reviewer raising the issue regarding the moderate agreement of Gemini-2.5-Flash. As also noted by Reviewer PjAS, relying on a single judge model can introduce bias. **Therefore, we conducted a comprehensive multi-judge study using three distinct architectures: (1) Gemini-2.5-Flash, (2) Qwen3-235B-A22B, and (3) Grok-4-Fast.**
>
>
> Results:
>
>
> 1. **Agreement with Humans**: We found that Gemini and Qwen offer comparable agreement with human annotators, while Grok performs better overall.
>
>
> 2. **Cross-Judge Agreement**: We observed high inter-annotator agreement among the three LLM judges across the full dataset.
>
>
> 3. **Self-Preference Bias**: To test for bias, we measured cross-judge agreement on CoT traces generated by Gemini. We compared agreement pairs: (a) Qwen-Grok, (b) Gemini-Qwen, and (c) Gemini-Grok. If Gemini were biased toward its own outputs, we would expect lower agreement between it and the other judges. However, the Gemini–Qwen pair exhibited the highest agreement, suggesting that Gemini remains objective even when evaluating its own generations in this setup.
> Full details of this analysis have been added to Appendix H (starting at line 1170).
>
> **W3+Q2: Comparison to Existing Methods**
> We agree that a quantitative comparison to existing reasoning metrics is essential.
> Following standard evaluation protocols [1, 2], **we calculated the correlation (Somers’ D [3]) between the scores derived from different evaluation approaches versus human quality judgments** (1–5 Likert scale) on the e-SNLI dataset [4].
> The results are: ROSCOE [1]: 0.30,  SocREval [2] (Base): 0.30,  SocREval [2] (Prompt Engineered): 0.58,  Our Approach (Average of metrics): 0.45.  (ReCEval does not provide compatible metric values for this comparison.)
> While our method focuses on granular diagnosis rather than a single holistic quality score, it demonstrates a clear improvement over the base implementations of ROSCOE and SocREval. While heavily prompt-engineered SocREval achieves higher correlation, our method provides granular diagnostic insights that holistic scores (i.e. indicating `how “good” is the explanation’) cannot capture. We added this empirical comparison in lines 1341-1349. Additionally, we will include a comparison of the correlation between the existing evaluation metrics and the final entailment accuracy in the camera-ready version. In addition to the empirical results, **our method provides an evaluation** for the individual steps involved in NLI reasoning - **allowing researchers to understand in which reasoning stages the models get wrong**, and accordingly to direct research at improving the weaker aspects in the reasoning process.
>
> **W4a: Scope**
> We acknowledge that Fact Verification and NLI represent a specific subdomain within the broad scope of ICLR. However, these tasks are foundational to the critical challenge of hallucination mitigation -- a problem that currently drives a vast body of research [5-7]. **Since NLI is a primary mechanism for ensuring model faithfulness, we believe our contributions offer diagnostic insights applicable to the wider field of reliable text generation.** Hence, we believe that focusing a research on evaluating the specific components of NLI reasoning is of sufficient importance.

---

> > ### Author Response · Authors · 2025-11-21
> >
> > **W4b: Attribution Typology**
> > We agree that the distinction between attribution types (extractive vs. abstract) was underutilized in our experiments. We intended this typology to provide further insights on the nature of attributions, rather than considering it as an evaluation factor (since either type of attribution may be legitimate). Though, we agree that there is room to further investigate the potential utility and quality of these types in future research. Specifically, localized (sentence-level) attribution is often more actionable for users than abstract (page-level) attribution. We will clarify in the final version.
> >
> > [1] ROSCOE: A SUITE OF METRICS FOR SCORING STEP-BY-STEP REASONING; Golovneva et al., 2023
> >
> >
> > [2] SOCREVAL: Large Language Models with the Socratic Method for Reference-Free Reasoning Evaluation; He et al., 2024
> >
> >
> > [3] A new asymmetric measure of association for ordinal variables; Robert H Somers, 1962
> >
> >
> > [4] A large annotated corpus for learning natural language inference; Bowman et al., 2015
> >
> >
> > [5] A Survey on Hallucination in Large Language Models: Principles, Taxonomy, Challenges, and Open Questions’ Huang et al., 2023
> >
> >
> > [6] A Comprehensive Survey of Hallucination Mitigation Techniques in Large Language Models; Tonmoy et al., 2024
> >
> >
> > [7] Large Language Models Hallucination: A Comprehensive Survey; Alansari et al., 2025

---

> > > ### Author Response · Authors · 2025-11-27
> > >
> > > Dear Reviewer,
> > >
> > > thank you for your review and for considering our paper. We've added a response to your comments; with the rebuttal deadline approaching, we'd be very grateful if you could take a quick look and share any final thoughts or remaining concerns. Thanks again!

---

### Official Review · Reviewer_PjAS · 2025-10-31

**Soundness:** 2
**Presentation:** 3
**Contribution:** 2
**Rating:** 4
**Confidence:** 3

**Summary:**

This work proposes a type-aware evaluation framework for NLI-style fact verification, breaking models’ reasoning traces into decomposition, attribution, entailment, and aggregation, scoring each for existence and quality. They annotate 360 model outputs sampled from ClearFacts, and LLM-as-a-judge after demonstrating high inter-annotator agreement, finding that models often skip steps, the correctness of entailment correlates strongly with overall accuracy, that guided prompting can narrow the gap with reasoning models, and non-reasoning models with guided CoT can be more token-efficient.

**Strengths:**

1. The clear separation of decomposition, attribution, entailment, and aggregation pairs well with the manual evaluation on a more principled and granular lens, and the LLM-as-a-judge helps to scale with substantial/almost-perfect agreement on the majority of metrics.
2. The inclusion of a human study with reported IAA is valuable, using Gwet’s AC, with substantial or almost-perfect agreement, and the correlation analysis over the attributes provides important insights.
3. The finding that extractive attribution is most reliable, followed closely by paraphrase attribution, before a major drop-off with abstract attribution, is intuitive and actionable.

**Weaknesses:**

1. While the LLM-as-a-judge is effective along many lens, it only has moderate agreement for entailment and inference, which are described to be the most important quality metrics, and the LLM-as-a-judge findings drive several of the conclusions made.
2. The aggregation logic is a bit simple, as there may be other aggregation operators that do not necessarily require all sub-claims to be obviously entailed, but can derive compositional value.
3. Decomposition is binary per trace, but other existence metrics are defined by proportions over sub-claims, and since guided prompting increases sub-claim count, this can make later “existence” values harsher for comparison.
4. One concern is that the judge is Gemini-2.5-Flash, while Gemini variants are also one of the generators, so it’s unclear if there may be some bias present.

**Questions:**

1. Do your conclusions change if you normalize existence/quality by token budget or by a fixed number of sub-claims?
2. Can you replicate the scaled up results with multiple judges and report cross-judge reliability, especially for inference and entailment?
3. In l268 and l272, please do not have the subscript be the remaining characters — instead, spell out the full word “infer” and “entail” in the subscript, while the value can still be denoted by I and E, respectively, as they currently are.

---

> ### Author Response · Authors · 2025-11-21
>
> We thank the reviewer for the thoughtful and detailed feedback. We are glad to see that the core contributions -- the proposed evaluation protocol, and the manual annotations -- were found valuable. Below we address each comment. Given the space constraints, we have detailed the new results in the revised manuscript. We explicitly indicate the specific section or table where each issue is addressed.
>
> **1. Moderate Results for LLM-as-a-Judge**
> We agree that achieving high agreement on critical metrics like entailment and inference is essential for the validity of our automated evaluation.
> To address this, we conducted a new **LLM-as-a-Judge evaluation using Grok-4-Fast. This stronger model yields 'Substantial' or 'Almost Perfect' agreement** with human annotators across all metrics (e.g., inference and entailment metrics get a Gwet’s score of 0.64 and 0.65, respectively. Granularity and completeness quality metrics both get a score of 0.87 — above the human agreement. Additionally, the overall decision metric gets a Gwet’s score of 0.78). Importantly, all empirical trends reported in the original paper remain consistent under this improved judging setup. These updated results are included in the revision (lines 447-450 and line 940-table 5).
>
> **2. Aggregation Logic and Alternative Operators**
> We agree that more expressive aggregation strategies may be informative beyond the strict operator explored in the main paper.
> **We therefore evaluated two additional aggregation prompts** across four models (Llama-3-70B, Gemini-2.0-Flash, DeepSeek-R1-Distill, Gemini-2.5-Flash):
>
>
> 1. Calculate the average entailment score of all the sub-claims, and classify as ‘faithfull’ if the average score is above 0.5 (inspired by [1])
>
>
> 2. Calculate the average contradiction score of all the sub-claims, and classify as ‘faithfull’ if the average score is below 0.5 (inspired by [2])
>
>
> Together with our unguided and guided CoT prompts, **this yields four aggregation strategies**.
> As shown in Table 11 (line 1135) of the revised paper, our guided CoT approach achieves the highest aggregation accuracy and the highest overall prediction accuracy across all models, which is reasonable, since a faithful statement requires all its sub-claims to be fully entailed. The unguided CoT gets the second place, the the entailment-based score and in the last place is the contraidted-base score.
>
> **3. Sub-claim–based Existence Metrics and Comparability**
> We agree that differences in the number of sub-claims may complicate comparisons if not normalized correctly. **There are two common ways to compute these metrics**:
>
>
> - Macro-average: Calculating the proportion of sub-claims per instance, and then averaging these instance-level scores.
>
>
> - Micro-average: Calculating the proportion of total generated sub-claims across the entire dataset that possess a component.
>
>
> In our setup, **we adopt the macro-average approach**, as it better reflects the model’s typical behavior on a per-statement basis and avoids overweighting statements that produce larger decompositions. Further, since the existence measures are computed as a proportion per instance, it normalizes across different methods whose decomposition may produce a different number of sub-facts per instance.
>
> **4+Q2. LLM-as-a-Judge**
> We thank the reviewer for raising this point.
> **We expanded the LLM-as-a-Judge study using three judges**: (1) Gemini-2.5-Flash, (2) Qwen3-235B-A22B, and (3) Grok-4-Fast
>
> We evaluated:
>
>
> 1. **Agreement with manual annotations**, showing that Gemini and Qwen are comparable, and Grok performs better and more reliable performance.
>
> 2. **Cross-judge agreement** over the entire dataset (excluding each model’s own generations), finding high agreement among all three judges.
>
> 3. **Potential judge–generator bias:** for Gemini-produced CoTs, we measured agreement across (a) Qwen–Grok, (b) Gemini–Qwen, and (c) Gemini–Grok.
> If Gemini were biased toward its own outputs, we would expect lower cross-judge agreement. Instead, Gemini–Qwen exhibits the highest agreement, suggesting LLMs might be objective judges at this setup.
>
> Full results are provided in Appendix H (starting at line 1170).
>
> **Q1: Analysis Normalization**
> We thank the reviewer for this suggestion.
> To determine if our conclusions are correlated by generation budget or granularity, **we analyzed the correlation between our quality metrics and three parameters: (i) token count, (ii) generation cost, and (iii) number of sub-claims.**
> We fitted linear regression models for each quality metric against these parameters.
> As shown in Appendix I (table 19, line 1315) the regression coefficients are consistently small and all p-values are high (p > 0.1). This indicates that none of the examined factors - token count, cost, or decomposition granularity - has a statistically significant relationship with any of the quality metrics.

---

> > ### Author Response · Authors · 2025-11-21
> >
> > **Q3:** We have updated the notation.
> >
> > [1] SummaC: Re-Visiting NLI-based Models for Inconsistency Detection in Summarization; Labban et al, 2021
> > [2] Explainable Hallucination through Natural Language Inference Mapping; Chen et al, 2025

---

> > > ### Author Response · Authors · 2025-11-27
> > >
> > > Dear Reviewer,
> > >
> > > We've responded to your review with clarifications and new results. As the rebuttal deadline is approaching, we'd be grateful if you could revisit our response and provide any final feedback. We appreciate your consideration.

---

### Official Review · Reviewer_qvPw · 2025-11-01

**Soundness:** 2
**Presentation:** 2
**Contribution:** 3
**Rating:** 4
**Confidence:** 3

**Summary:**

This paper proposes a framework for evaluating how LLMs reason during fact verification. The authors argue that current evaluation methods are too general, so they design a structured approach that divides reasoning into four stages: decomposition, attribution, entailment, and aggregation. They test various LLMs, comparing unguided and guided reasoning prompts, and find that models often skip key reasoning steps unless explicitly guided. Guided prompts generally improve reasoning completeness and accuracy, especially for smaller or non-reasoning models. The study shows a strong link between reasoning quality and correct predictions, and suggests that structured evaluation can help diagnose weaknesses and improve both reasoning quality and reliability of LLM outputs.

**Strengths:**

1. The authors frame fact verification as a reasoning task and define a diverse set of reasoning components, which provides a valuable perspective for redefining the fact verification task in reasoning models.

2. The individual reasoning components proposed by this paper, as well as the overall system design for fact verification, are reasonable.

3. By involving human annotators, the authors increased the reliability of the validation of their proposed methodology.

**Weaknesses:**

1. Overall, the authors decompose the fact verification task into multiple components and define it compositionally. The structure of each component and the overall process (as suggested in Figure 2) sounds reasonable. However, in order to demonstrate that this actually contributes to LLM fact verification, the paper should have provided broad analysis across a variety of models (different model families and capacities) and across multiple benchmarks. In that respect:

  1-1. The authors claim that applying their method to non-reasoning models and using guided CoT can yield performance gains, but experiments were conducted on a very limited set of models (Llama and Gemini 2.0 Flash).

  1-2. The ClearFacts benchmark is a good dataset for this task, but despite the existence of many other datasets for fact verification, the experiments were conducted on only a single benchmark.

  1-3. A total of 360 human judgments is not enough to establish the reliability of the human annotations.

2. The component taxonomy presented in the paper is reasonable, but the methodology itself is not novel. It essentially compares results before and after prompting. I’m not opposed to influencing model behavior through prompting, but it’s disappointing that the proposed complex system (which aims to perform multiple NLI tasks simultaneously) is implemented with just a single prompt. A more fine-grained, component-level analysis could have been done.

3. The proposed approach appears to rely heavily on decomposition. The paper should include more qualitative examples showing how decomposition is performed for different statements.

**Questions:**

Questions and suggestions are listed in Weakness section.

---

> ### Author Response · Authors · 2025-11-21
>
> We thank the reviewer for their thoughtful feedback and for recognizing the value of our reasoning-based perspective on fact verification. We also appreciate the validation of our component design and the reliability of our manual annotation process. We have addressed the specific concerns regarding experimental breadth and qualitative analysis below. Due to space limitations, we have included the full results in the revised paper rather than here. However, we reference the specific sections and tables where these findings are presented.
>
> **1-1. Number of Evaluated Models**
> We acknowledge that broadening the model selection strengthens the generalizability of our claims. In the revised version,**we have expanded our experiments to include three additional strong non-reasoning models**: Qwen3-30B-A3B-Instruct, Qwen3-235B-A22B-Instruct, and Grok-4-Fast (the non-reasoning variants). See line 304 and the tables in lines 975 and 990.
>
> Across these new models, we observe the same trends reported in the original submission, including:
> (1) guided CoT improves the performance of non-reasoning models, and
> (2) stronger models achieve higher scores across the proposed reasoning components.
> These results significantly reinforce the robustness of our findings.
>
> **1-2. Use of a Single Benchmark**
> Although we agree that extending to additional datasets would strengthen the work, we clarify that **ClearFacts is itself a composite benchmark, constructed from 14 distinct existing fact-verification datasets** spanning older and more recent sources (including AggreFact-CNN, AggreFact-XSum, ClaimVerify, ExpertQA, FactCheck-GPT, LFQA, RAGTruth, Reveal, TofuEval-MediaS, TofuEval-MeetB, Wice, CoverBench, Hover and SciFact) - we added a clarification in table 20 (line 1350).
> Because of its breadth and diversity, ClearFacts offers robust coverage of common fact-verification phenomena. We therefore believe it provides the necessary breadth and representativeness for a robust compositional evaluation.
>
> **1-3. Number of Human Annotations**
> We agree that increasing the diversity of source statements improves statistical reliability.
> To improve reliability, **we have already begun annotating more unique statements -- we will at least double the size of the human-annotated dataset, resulting in more than 720 total reasoning samples.** This expanded annotation set will be included in the camera-ready version.
> Initial results from the new samples confirm the trends reported in the submitted paper, providing further statistical support for our conclusions.
>
> **2. Novelty and Prompting Methodology**
> We appreciate the opportunity to clarify the scope of our contribution. Our goal is not to propose a new methodology for fact verification. Rather, our main goal was to introduce a novel evaluation methodlogy for the quality of the reasoning provided by LLMs for the fundamental NLI task, which is the underlying reasoning process for fact checking. Then, we aimed to apply this methodlogy to evaluate the most common scenario in which LLMs are emploeyd to perform NLI inference. Indeed, **many users prefer single-prompt approaches** for their simplicity, ease of deployment, evaluation, and reproducibility. **Our aim is therefore to rigorously analyze this most common paradigm.**
> In this regard, our work aligns conceptually with ROSCOE (ICLR 2023) and OCEAN (ICLR 2025), which also focus on a novel evaluation methodology.
>
> **3. Qualitative Examples of Decomposition**
> We agree that concrete examples are essential for intuitively understanding the decomposition mechanics. **We have added a set of qualitative examples to the revised paper** (now presented in Table 10, line 1080). These examples cover a broader range of statement types to clearly demonstrate how the decomposition process operates in practice across different contexts. We observe that there is no single consistent structure for the decomposition outputs. This suggests that the decomposition step enables further investigation.
>
> We thank the reviewer again for their helpful comments, which have motivated us to significantly strengthen our work.

---

> > ### Author Response · Authors · 2025-11-27
> >
> > Dear Reviewer,
> > Since the rebuttal period is close to ending, we wanted to kindly remind you to review our comments and let us know if anything is still unclear or needs further clarification. Many thanks for your time.

---

### Meta-Review · Area_Chair_jJjP · 2026-01-07

**Summary:**

This paper investigates LLM reasoning quality during fact verification. Compared with existing generic evaluation, this paper offers a systematic and fine-grained evaluation based on four aspects: decomposition, attribution, entailment and aggregation. The introduced protocol evaluates both existence and quality for each aspect. Evaluations on multiple LLMs using the proposed protocol reveal some interesting findings, such as positive correlation between reasoning trace quality and final prediction accuracy, model's tendency of skipping certain reasoning steps, effectiveness of guided prompt for reasoning quality and final performance.

Overall, this paper offers a systematic analysis with fine-grained evaluations on the reasoning quality of LLMs for fact verification. The decoupled protocol provides better diagnosis on model's capability. The analysis and findings are interesting, and offers new insights for improving model reasoning and task performance.

Meanwhile, the reviewers agree on the following weaknesses that deserve further investigations and revisions:
- The size of the evaluated data is too small, and thus weakens the findings. The data is composed of 30 samples which is far below the normal scale.
- The evaluation framework relies on LLM-as-a-judge for entailment and inference, where many LLMs are not so good at yet. This makes the automated framework less reliable and sensitive to the choice of LLMs. The lack of statistical rigor for cross-judge reliability, especially for inference and entailment, is an issue.
- The framework lacks generalizability towards other reasoning tasks beyond fact verification.
- The framework is not novel, as methods like ROSCOE also introduces similar fine-grained diagnosis.
- The findings in this paper are not quite surprising. Even though the rebuttal has revealed some interesting findings, these are not orginally intended in this work. The authors are encouraged to re-structure or re-frame their claims in the paper, considering the new insights identified during the rebuttal phase.

**Reviewer Concerns:**

Concerns being addressed:
- The scalability of experiments including more LLM families and clarification on the benchmark coverage.
- More qualitative examples of the decomposition has been added.
- More aggregation methods are tested which provide a more comprehensive comparison.
- Testing with other LLMs as judges and discussing the potential bias issue.
- Comparison with some baselines such as ROSCOE

Outstanding concerns:
- The size of the evaluated data is small, and thus weakens the findings. The data is composed of 30 samples which is far below the normal scale.
- Reliance on LLM-as-a-judge which does not show strong performance for certain LLMs.
- Statistical rigor for cross-judge reliability, especially for inference and entailment.
- The framework lacks generalizability towards other reasoning tasks beyond fact verification.
- The framework is not novel, as methods like ROSCOE also introduces similar fine-grained diagnosis.
- The findings in this paper are not quite surprising. Even though the rebuttal has revealed some interesting findings, these are not orginally intended in this work. The authors are encouraged to re-structure or re-frame their claims in the paper, considering the new insights identified during the rebuttal phase.

**Reviewer Scores:**

I don't think the reviewer would have changed their score if more discussions are involved.

---

### Decision · Program_Chairs · 2026-01-26

Reject